# Proof of Concept of the Contribution of the Interaction between Trait-like and State-like Effects in Identifying Individual-Specific Mechanisms of Action in Biological Psychiatry

**DOI:** 10.3390/jpm12081197

**Published:** 2022-07-23

**Authors:** Sigal Zilcha-Mano, Nili Solomonov, Jonathan E. Posner, Steven P. Roose, Bret R. Rutherford

**Affiliations:** 1Department of Psychology, University of Haifa, Mount Carmel, Haifa 31905, Israel; 2Weill Cornell Institute of Geriatric Psychiatry, Weill Cornell Medicine, White Plains, NY 10605, USA; nis2051@med.cornell.edu; 3New York State Psychiatric Institute, Columbia University Vagelos College of Physicians and Surgeons, New York, NY 10032, USA; jonathan.posner@nyspi.columbia.edu (J.E.P.); spr2@cumc.columbia.edu (S.P.R.); brr8@cumc.columbia.edu (B.R.R.)

**Keywords:** mechanisms of action, between-individuals variance, within-individual variance, precision medicine, mediators, moderators

## Abstract

**Background:** Identifying individual-specific mechanisms of action may facilitate progress toward precision medicine. Most studies seeking to identify mechanisms of action collapse together two distinct components: pre-treatment trait-like characteristics differentiating between individuals and state-like characteristics changing within each individual over the course of treatment. We suggest a conceptual framework highlighting the importance of studying interactions between trait-like and state-like components in the development of moderated mediation models that can guide personalized targeted interventions. **Methods:** To facilitate implementation of this framework, two empirical demonstrations are presented from a recent clinical trial and neuroimaging study. The first examines limbic reactivity during an emotional face task; the second concerns striatal activation in a monetary reward task. **Results:** In both tasks, considering the interaction between trait-like and state-like components predicted treatment outcome more robustly than did the trait-like or state-like components examined individually. **Conclusions:** These findings suggest that the extent to which state-like modulation of neural activations can serve as a potential treatment target depends on the pre-treatment, trait-like levels of activation in these regions. Thus, the interaction between trait-like and state-like components can serve as a promising path to the development of personalized interventions within a precision medicine framework in which mechanisms of action are individual-specific.

## 1. Introduction

Some studies in biological psychiatry have defined baseline trait-like patient characteristics that predict treatment response (biotypes and biomarkers), while others have identified putative therapeutic mechanisms (brain changes associated with treatment efficacy), but no research has connected these complementary information sources. Yet, the exploration of interactions between baseline trait-like patient characteristics and state-like changes of a construct has long been recognized as a promising pathway to understanding individual-level mechanisms in medicine. An early example is a classical editorial by Curfman [1] which integrated the findings from two large studies published in the New England Journal of Medicine on the effect of physical exertion on the risk of heart attacks. Curfman proposed that the adverse effect of state-like (SL) changes (for example, increases) in the level of exercise is protected (moderated) by the regular trait-like (TL) levels of exercise of the individual. Specifically, for individuals whose TL levels of exercise suggested that they exercised regularly, there was no effect of SL increases in the level of exercise on the risk of heart attack. By contrast, for individuals whose TL levels of exercise suggested that they did not exercise regularly, an SL increase in exercise resulted in greater risk of heart attack [2,3].

The focus on the interaction between the TL and SL components of exercise and the risk of heart attack provides an elegant explanation of the mixed results reported in the literature, where the TL effect of exercise on heart attack was positive (namely, individuals who exercise regularly show lower risk of heart attack), whereas the SL effect of exercise on heart attack was negative (namely, when an individual increases the level of exercise, the risk of a heart attack increases). The interaction between TL and SL components means that regular exercise (the TL effect) protects against the triggering effect of strenuous exertion (the SL effect) (Mittleman et al., 1993) [2]. The complete picture of the effects of exercise on the heart could emerge only when considering the interaction of both SL and TL components. 

The TL–SL distinction is especially important in the context of the therapeutic aim of normalizing pre-treatment abnormalities. The TL component signals the individual’s abnormalities, and the SL changes indicate how these abnormalities can be normalized (Figure 1). Our focus in this article is on the SL × TL interaction as the main path to precision psychiatry, in which variability between patients is utilized to identify the most effective treatment for each individual [4] We first describe the importance of the SL–TL distinction using an example from psychotherapy research. Second, we demonstrate the potential of investigating the SL × TL interaction in neurobiological studies in psychiatry. 

## 2. The Utility of the SL–TL Distinction

Distinguishing between SL and TL effects, including by measurement taken over time on multiple levels of analysis (i.e., self-report, behavior, physiology, circuits, cells, molecules, and genes), is particularly important in the study of mechanisms of action in psychiatry. Identifying the between-individuals variability of a given construct (the TL component) can help us personalize certain treatment components to given individuals. TL refers to any variability in individuals’ traits or stable pre-treatment characteristics. TL components may serve as (a) *prognostic* variables or predictors—strengths or deficits that may affect the patient’s ability to benefit from *any* treatment (e.g., cognitive impairment, interpersonal pathology, or social support)—and (b) *prescriptive* variables or moderators—strengths or deficits that may affect the patient’s potential benefit from a certain optimal intervention compared to alternative treatments. As illustrated in Figure 1, identifying the TL components (e.g., biomarkers, biotypes) can inform clinical decisions by tailoring a treatment to target SL components that are relevant to the patient’s specific needs and baseline characteristics. 

Understanding the effect of the SL component (within-individual variability) on candidate mechanisms of action can inform ways to target these mechanisms with selected interventions. *States* refer to any changes in patient characteristics over the course of treatment. These include both changes that are the result of treatment (e.g., cognition, social functioning) and those that result from events outside of treatment (e.g., stressful life events). By disentangling SL from TL effects, we can identify SL-level changes over time in individuals’ neurobiology (e.g., brain changes) and behavior (e.g., cognitive or social functioning). Based on these nuanced observations, we can gain knowledge about promising mediators or mechanisms of action that can serve as treatment targets in specific interventions.

## 3. The Utility of the SL–TL Distinction in Psychotherapy Research: The Case of the Therapeutic Alliance

The therapeutic alliance between the patient and the therapist is one of the most consistent predictors of outcome in psychotherapy research. A recent meta-analysis based on 295 studies suggested that a stronger alliance is a significant predictor of better treatment outcome across treatment orientations and patient populations [5,6] This finding may be an indication of a robust TL effect, where, on average, patients who form a stronger alliance also exhibit greater treatment response. However, such TL finding does not translate directly into clinical practice so that therapists will be able to improve the alliance with individual patients to facilitate therapeutic change [7,8,9,10,11]. Contemporary theoretical conceptualizations suggest that the effect of the alliance varies with different subpopulations of patients and across different treatments. For example, whereas in alliance-focused treatments, the alliance is conceptualized as a main mechanism of action, in standard cognitive behavioral protocols, it is perceived as a common non-specific factor that augments the efficacy of prescribed [12,13].

The SL–TL distinction enables researchers to capture more accurately the variation in the therapeutic effect of the alliance across individuals and treatments and translate findings into clinical recommendations [8]. Consider two patients with an equal alliance score on the third session of treatment. The patients may or may not differ in their TL potential to form a strong alliance. For example, one patient may have entered treatment with better interpersonal skills and more positive expectations from the alliance with the therapist than the other patient. Similarly, the two patients may or may not differ in their SL alliance. Their trajectories of change in alliance levels (fluctuations) over treatment may be different or similar. To understand how to improve the alliance and, in turn, the treatment outcome of these two patients, we need to measure both their TL tendencies and the SL changes occurring over the course of treatment. For example, a patient who shows a poor TL alliance with SL reductions in reported quality of the alliance may require more targeted interventions focused on addressing negative perceptions of the therapist and/or treatment compared to a patient with a stronger TL alliance and fewer SL fluctuations in the alliance. Thus, focusing on the interaction between the variability in trajectories of SL change over time and patients’ stable individual traits can help therapists devise personalized interventions. 

Studies that implemented the TL–SL distinction have produced important findings about the role of alliance in different treatments for various patients (for a review, see [14]. First, findings support the assumption that patients differ significantly in their TL potential to develop a strong alliance and that these differences are a product of their baseline characteristics, especially interpersonal characteristics [15,16,17,18,19,20,21]. Second, studies support the assumption that patients differ significantly in their SL alliance, with different patients showing distinct trajectories of SL fluctuations in the quality of alliance during treatment [22,23,24,25]. Third, previous work also demonstrated that the role of alliance may vary across individuals and treatments. For example, consistent with theoretical conceptualizations, studies suggest a stronger effect of SL alliance on treatment outcome in treatments where changes in alliance are conceptualized as the main mechanism of change and in the case of patients with poorer TL alliance [26,27,28].

The TL component of the alliance reflects the general potential of individuals to form a strong alliance with their therapist. Therefore, it can play the role of a *prognostic* variable (one’s ability to benefit from treatment in general) or a *prescriptive* one (facilitates the efficacy of one treatment more than others, such as alliance-focused treatment vs. standard cognitive behavioral therapy). By contrast, the SL component reflects changes in alliance that occur during the course of treatment and operates as a mechanism of action or a mediator that may lead to greater treatment response in psychotherapy [11] or psychopharmacological treatment [29]. However, alliance is only one example of the important potential contribution of the SL–TL distinction to psychotherapy research. The possibilities are vast, especially for constructs in which, unlike in the alliance, the SL and TL effects are expected to change in opposite directions over time [9]. In these cases, the SL–TL distinction can potentially resolve inconsistent findings, as in the example of the effect of exercise on heart attacks.

## 4. How to Implement the SL–TL Distinction in Clinical Research?

**Study design**. At the design level, repeated measures should be implemented, at least at three time points, preferably more. The exact number of assessment points required may differ based on the nature of the SL and TL components of the construct. To accurately capture the TL component of each construct, it is necessary to take into account the nature of the construct. The most important question is whether the TL component has a temporal pattern. Some traits have no temporal pattern (are not expected to change over time, such as history of childhood adversity; Medeiros et al. [30] and can be accurately estimated based on a single assessment, as they have no potential to change, and there is no error in their estimation. Some demographic variables are of this type, for example, age and sex as a biological variable. Other constructs have no temporal pattern, but their assessment involves errors, and therefore, multiple assessments are needed to accurately estimate the TL component of the construct. For example, structural brain profile, a potential TL component, changes slowly over the course of the lifespan, but given varying signal-to-noise ratios during MRI scans, it is important to measure it more than once at baseline if feasible. Other TL constructs do have a temporal pattern; that is, they are dynamic trait characteristics; when estimating these TL components, it is critical to take into account the temporal dynamics. In these cases, it is important to capture the dynamic TL patterns of the construct before the start of the intervention [31]. When a TL dynamic is present in the construct of interest, the SL change may manifest as a change in the construct mean or slope above the TL pattern, as a change in the TL pattern, or both.

To accurately capture the SL component, the therapeutic construct of interest should be sampled with a frequency that depends on the expected rate of change in that construct. The number of time points and the required time resolution (moments in a session, segments of a few minutes in each session, between-sessions assessments, etc.) needed to accurately capture the SL component of a given construct depend on its nature. Specifically, if the rate of change is slower (such as in structural brain profile), less frequent assessments are sufficient, but if the rate of change is high, session-by-session assessment or even within-session assessment is necessary. For example, resting-state functional connectivity can change significantly as a result of intervention. Yet, many current studies include only a baseline measurement, which limits the ability to study temporal fluctuations over treatment. If the construct of interest is expected to change in the time that elapses between the therapy sessions and not only during the sessions themselves, repeated measures using experience sampling should be considered as well.

**Statistical analyses**. Several statistical methods, including centering and detrending of the variable of interest, are available for disentangling between-individuals and within-individual effects in longitudinal studies [32,33]. Centering refers to the subtraction of a given data point from individuals’ overall mean or pre-treatment score. Detrending involves mathematically removing the time trend from a time series (e.g., removing the general line of improvement the individual shows in the course of treatment). When testing the effect of a predictor measured repeatedly during the intervention on outcome, detrending controls for the effect of time. The decision of which method to choose should be based on theoretical assumptions regarding the definition of *therapeutic change* and the research questions posed. For example, if a researcher is interested in change from baseline, a clear definition of baseline should be determined (e.g., one measurement pre-treatment vs. repeated daily measurement in the month before the start of treatment). Such decisions should be made according to the temporal pre-treatment patterns of the construct. As noted above, TL characteristics that are not expected to show fluctuations in the absence of intervention can be measured once, whereas TL characteristics that show inherent fluctuation even without intervention, such as negative affect, should be measured repeatedly at baseline. The definition of baseline also determines the way change from baseline is assessed. If the baseline is defined as the pre-treatment level, then SL is defined as the deviation from it. If the baseline is defined as the dynamic tendency estimate during the weeks before the treatment, then SL is defined as the deviation from the individual’s dynamic tendencies.

## 5. SL–TL Distinction in Biological Psychiatry

Traditionally, in psychiatric trials, the terms “trait” and “state” referred to Axis II and Axis I disorders, respectively. This classic view perceived trait and state as mutually exclusive and relating to entirely different constructs. Thus, any given construct can presumably serve either as a state *or* as a trait: personality constructs are referred to as trait; by contrast, mood and mood disorders are referred to as state [34]. However, this distinction appears to be only partially accurate. For instance, traditionally, personality disorders (PDs) are perceived as a prototypical example of traits: by definition, they reflect enduring patterns of inner experience and behavior that deviate markedly from the expectations of the surrounding culture and are pervasive, inflexible, and stable over time [35]. However, empirical evidence from the last decade suggests a more complex picture: PDs have both trait-like and state-like components [36]. For example, a study that used a naturalistic daily diary, focusing on variability over time and situations in PDs, suggested that daily PD manifestations fluctuated considerably over days. The study also found that individuals differed not only in average levels of daily PD features but also in the extent to which these features showed variability over time [37]. The empirical literature demonstrates that both TL and SL components can be identified not only in constructs that have been traditionally referred to as traits but also in those that have been traditionally perceived as states, such as affective symptoms. For example, a distinction has been made between trait anxiety as a personal characteristic and state anxiety during a given time period [38]. 

Kraemer et al. [39] noted that two distinct components of the same construct can play different roles in treatment. For example, lack of social support at baseline differs from changes in social support during treatment. It can be argued that each construct in biological psychiatry also has both TL and SL components and that each component may play different roles in treatment. For example, it is possible to test the TL effect of inflammatory markers by examining whether individuals at higher risk of depression tend to show higher levels of inflammatory markers than those at lower risk. Answers to such questions may be instrumental in identifying individuals with a greater predisposition for depression. It is also possible to test the SL effect by investigating whether decrease in levels of inflammatory markers is followed by reduction in depressive severity. Identifying the causal relationships between a candidate mechanism and outcome can help in the development of targeted interventions, such as techniques to reduce inflammation pathways in depressed individuals. Another example is the study of frontal alpha asymmetry (FAA). The TL effects can be defined as the variability between depressed individuals and healthy controls and the SL effects as changes occurring in treatment responders [40]. Yet another example is dopaminergic and glutamatergic dysregulation in schizophrenia. The TL effects can be defined as characteristics differentiating between antipsychotic-naive individuals with 22q11 deletion and healthy volunteers [41]. 


**Demonstrating the utility of the SL × TL interaction in biological psychiatry.**


In the following two examples, we demonstrate the utility of the SL–TL distinction and the examination of the SL × TL interaction. We chose two neuroimaging tasks assigned to patients with major depressive disorder (MDD). The tasks are commonly used in biological psychiatry to investigate amygdala activation during facial recognition and pallidum activation during anticipation and receipt of monetary rewards and penalties. The two examples are derived from a randomized controlled trial (RCT) in which patients with MDD underwent fMRI scans twice before the start of treatment, 1 week apart, and pre- and post-manipulation [42]. In these two examples, we expected the interaction between SL and TL components to reveal novel information that can govern future precision medicine, indicating who are the individuals who may benefit most from targeting changes in amygdala and pallidum activation. 

**Participants:** The study was conducted at the Clinic for Aging, Anxiety, and Mood Disorders at the New York State Psychiatric Institute (NYSPI). All procedures were approved by the NYSPI IRB. Eligible participants were aged 24–65 years. They met Diagnostic and Statistical Manual IV (DSM-IV) criteria for non-psychotic MDD, had a 24-item HRSD score ≥ 16, were right-handed, gave informed consent, and complied with study procedures. 

**Design and Methods:** Study procedures were described in detail elsewhere [42]. Briefly, 50 patients were enrolled in an 8-week antidepressant clinical trial experimentally manipulating expectancy. At baseline, patients underwent initial evaluation to assess eligibility, had pre-randomization psychiatric symptoms measured, and underwent MRI scanning (Scan 1). Participants at baseline had what they perceived to be a 75% probability of receiving active antidepressant medication. Pre-randomization expectancy was measured with participants having this knowledge. Next, participants’ expectancy of improvement was experimentally manipulated through instructions to participants about the probability of receiving active medication compared to receiving placebo. Participants were randomly assigned either to the placebo-controlled condition (50% chance of receiving active treatment) or to the open condition (100% chance of receiving active treatment) and informed of the results of the randomization. Post-randomization expectancy was measured with participants having this additional information (this was the expectancy manipulation). Those in the placebo-controlled condition were randomly assigned to medication or placebo. The second randomization within the placebo-controlled group was masked, and neither participants nor outcome assessors were aware of the randomization schedule or the specific treatment assignment to medication or placebo. fMRI Scan 2 was then performed within 1 week of the week 0 visit, after which either citalopram or a placebo pill was administered. Thus, both pre- and post-randomization outcome expectancy measurements and fMRI scans 1–2 were obtained before patients received any medication. HRSD was measured weekly over the 8-week clinical trial. Only individuals who had the two scans were included in the current analyses. Of the patients participating in the RCT, 23 met imaging criteria (no MRI-contraindications, etc.) and formed the effective sample for this secondary analysis. Of these patients, 9 were randomized to the open group and 14 to the placebo-controlled group (11 received medication, and 3 received placebo). No significant differences in demographic data or baseline clinical characteristics were found between participants who were and were not scanned or between participants in the placebo-controlled and open groups. For each individual patient, only two MRI scans were conducted in this trial. 

The first example concerns right amygdala activation in fear vs. neutral faces, and it was based on the masked emotional face task [43]; the second concerns left pallidum activation in response to reward cues, and it was based on the monetary incentive delay task (MIDT; Pizzagalli et al. [44]. Details on the design, measures, and methods appear in the Appendix A. Images were obtained on a GE Signa 3.0T whole-body scanner (Milwaukee, WI, USA) operating the E2-M4 platform and using a quadrature head coil in receive mode. SPM8 (http://www.fil.ion.ucl.ac.uk/spm/ (accessed on 6 March 2017) under MATLAB 2014B was used to preprocess the functional imaging data. Details of the image acquisition, pre-processing, and analyses appear in the Appendix A. 

**Results**: The analyses were conducted using the SAS PROC MIXED procedure for multilevel modeling [45]. We conducted two separate models, one for each construct: amygdala activation and pallidum activation. We referred to the baseline pre-manipulation levels of the two constructs as the “TL components” and to changes from baseline pre-manipulation to post-manipulation as the “SL changes”. The two examples illustrate a simple pre-post design commonly used in neurobiological studies in psychiatry. Repeated measurement designs with a high number of data points are more appropriate for establishing stable estimates of the SL and TL components. 

First, we evaluated the effect size of the TL and SL components of the two constructs on the slope of change in treatment outcome. We did so by evaluating the interaction of the TL component with log of time and the SL component in predicting HRSD levels repeatedly from baseline to post-treatment for each construct separately. We chose the scaling of time in log because it showed the best fit to the data. Second, we evaluated the effect size of the interaction between the TL and SL components on the slope of change in treatment outcome by testing the 3-way interaction between the TL, SL, and log of time in predicting HRSD levels repeatedly from baseline to post-treatment. We created separate models for each construct (right amygdala activation in fear vs. neutral faces; left pallidum activation in response to reward cues). We calculated effect sizes (f^2^) as described in [46].

For the right amygdala activation in fear vs. neutral faces, the effects of each of the two variables with the slope of change in HRSD were: for the TL (B = 0.52, SE = 0.12, *t*_(149)_ = 4.16, *p* < 0.0001, f^2^ = 13.3%) and for the SL (B = −0.34, SE = 0.12, *t*_(149)_ = −2.74, *p* = 0.007, f^2^ = 6.2%). The effect of the 3-way interaction was significant (B = −0.05, SE = 0.02, *t*_(148)_ = −2.09, *p* = 0.03, f^2^ = 2.9%; Figure 2). For the left pallidum activation in response to reward cues, the effects of each of the two variables with the slope of change in HRSD were not significant: for the TL (B = 0.33, SE = 0.71, *t*_(126)_ = 0.64, *p* = 0.64, f^2^ = 0.2%) and for the SL (B = 0.34, SE = 0.41, *t*_(126)_ = 0.83, *p* = 0.40, f^2^ = 1.5%). The effect of the 3-way interaction was marginally significant (B = −1.03, SE = 0.52, *t*_(125)_ = −1.97, *p* = 0.05, f^2^ = 3.1%; Figure 3). 

The two examples demonstrate the utility of the SL–TL distinction as well as the unique contribution of their interaction, the SL × TL. The TL component moderated the effect of the SL component on outcome so that the TL component determined the extent to which targeting SL changes resulted in improvement in treatment outcome and the direction of the changes that are required (increase vs. decrease). Regarding the amygdala, findings suggest that for individuals with low TL amygdala activation, regulation of the amygdala may not serve as a target of therapeutic interventions, whereas for those with high TL amygdala activation, targeting amygdala dysregulation may facilitate greater treatment response. Regarding the pallidum, the evidence suggests that some individuals with MDD show hyperactivation and others hypo-activation in this area [47]. In our small sample, the interaction was marginally significant, but the effect size was sizable, suggesting the potential importance of accounting for the TL levels.

Pending replication in a larger sample, our results suggest that patients with low TL activation in the pallidum may benefit from interventions targeting increase of activity in this region, while those with high TL activation may benefit from interventions targeting reduction of pallidum activity. Collecting further information about the patients’ other TL characteristics (in addition to amygdala and pallidum activation) can be instrumental in designing novel, tailored interventions for individual patients.

## 6. Summary and Discussion

A shift toward precision medicine in psychiatry requires identifying which *sub-populations* of patients respond optimally to which interventions. Such insights can be achieved by investigating the SL × TL interactions in moderated mediation models that can be translated into personalized clinical recommendations. These models can identify important moderators (TL, pretreatment patient characteristics) and mediators (SL, significant individual-level changes during treatment) and integrate their effects. Specifically, the TL component (moderator) indicates the subpopulation that may benefit most from a given target (mediation). Clinically, patients’ TL components could be assessed before the start of treatment, and then, an intervention could be tailored to fit each patient individually, aimed at optimizing TL abnormalities by inducing SL changes relevant to that patient. As illustrated in Figure 1, identifying the TL components that signal the individual’s abnormalities can help tailor interventions for individual patients. This framework takes into account the individual’s needs, clinical presentation, and circumstances. This method can also account for the heterogeneity in psychiatric conditions by finding the optimal treatment for each patient. This article offers a framework for identifying the mechanisms of action that need to be targeted for each individual but does not provide the details on which intervention may produce change in each mechanism.

The proposed framework concerning the potential of the SL–TL distinction for biological psychiatry research has important implications for future study design and analytic approaches. At the stage of trial design, researchers could identify potential TL components (candidate moderators) and SL components (mediators or mechanisms) of interest. These constructs can then be measured repeatedly over the course of the trial—before, during, and after treatment—to enable the estimation of both baseline levels (TL component) and trajectories of change over treatment (SL component). Treatment outcome should also be measured repeatedly and at the same time points to enable cross-lagged design testing of the temporal relationship between the mechanism of action and outcome. Similarly, adequate analytic methods for disentangling TL and SL effects should be implemented at the stage of statistical analyses.

In sum, the article illustrates the potential utility of the SL–TL distinction in biological psychiatry and its potential to further our understanding of personalized treatment models. We recognize that adopting this approach requires significant changes in both trial design and methodology and may present future challenges. Moving forward, we recommend an interdisciplinary collaboration between clinical researchers, data science experts, clinicians, and individuals with relevant lived experience in identifying candidate TL components that can moderate treatment response as well as central SL changes in mechanisms of action that can be targeted in both pharmacological and psychosocial interventions. This will enable: (a) crystalizing the most relevant constructs for investigation, (b) shedding light on their temporal characteristics when no intervention is provided as well as during effective intervention so that an appropriate measurement regime can be designed, and (c) choosing the most adequate data science approaches for disentangling TL and SL components that best fit the data and the conceptual model.

## Figures and Tables

**Figure 1 jpm-12-01197-f001:**
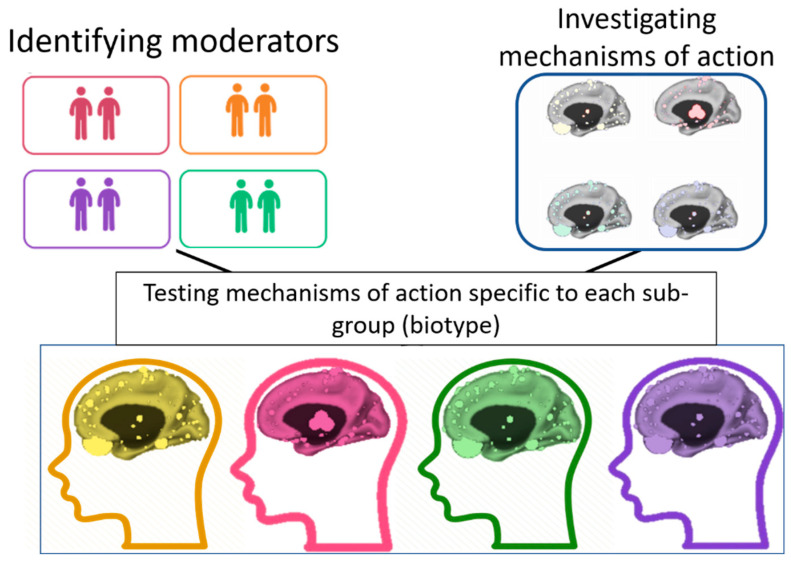
The study of moderators (trait-like characteristics; biotypes; **left**), the study of mechanisms of action (state-like changes; **right**), and their integration (**bottom**). Integrating the two fields of research of moderators and mechanisms of action is critical for identifying individual-specific mechanisms of action in the progress toward precision medicine.

**Figure 2 jpm-12-01197-f002:**
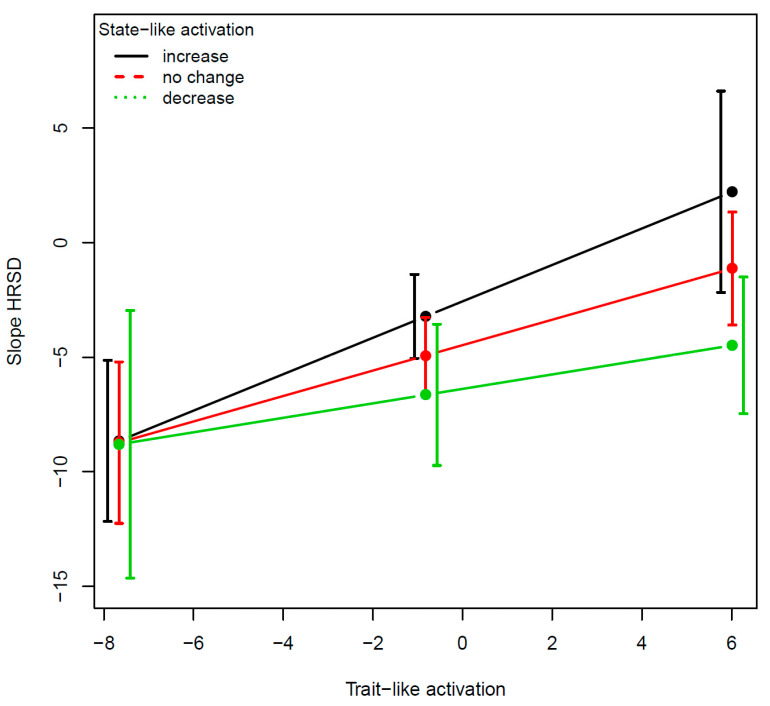
Example 1: SL–TL distinction in the case of right amygdala activation in fear vs. neutral faces. The figure is based on the predicted values from the models fitted on real data of actual patients who were enrolled in the RCT. The *y*-axis is the outcome variable defined as the predicted rate of HRSD change over time in units of log of week (the scaling of time, in log, was chosen according to the model that showed the optimal level of fit to the data). Lower levels on the *y*-axis refer to greater symptom reduction. The *x*-axis refers to the trait-like levels (baseline levels before the manipulation). The different lines refer to the state-like levels (amount of change that occurred from baseline to post-expectancy manipulation before the start of treatment). The black line indicates the increase in activation (one SD above no change in activation), the red line refers to no change in activation (about 39% of the patients showed less than one standard deviation away from 0 in the amount of change), and the green line refers to decrease in activation (one SD below no change in activation). The error bars represent 95% confidence intervals.

**Figure 3 jpm-12-01197-f003:**
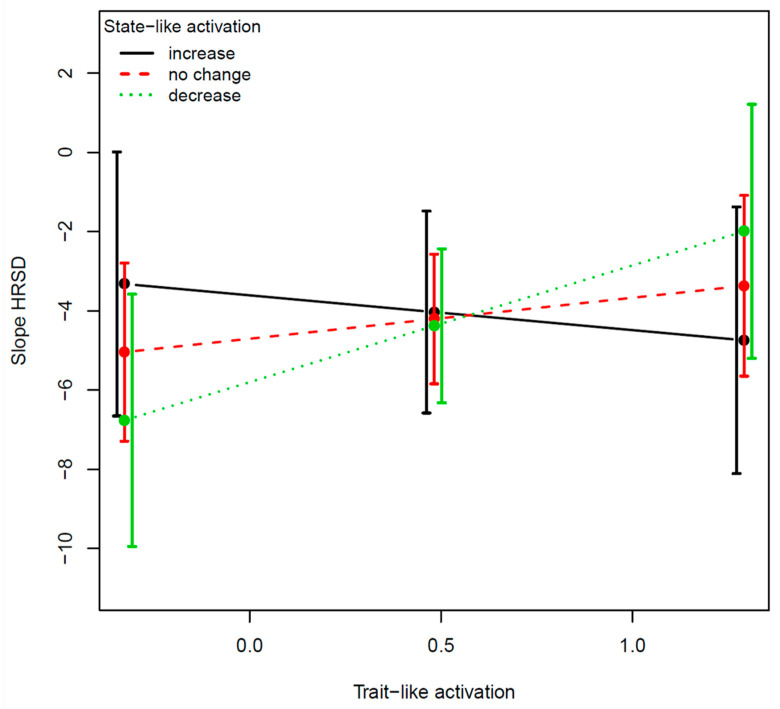
Example 2: SL–TL distinction in the case of left pallidum activation in response to reward cues. The figure is based on the predicted values from the models fitted on real data of actual patients who were enrolled in the RCT. The *y*-axis is the outcome variable defined as the predicted rate of HRSD change over time in units of log of week (the scaling of time, in log, was chosen according to the model that showed the optimal level of fit to the data). Lower levels on the *y*-axis refer to greater symptom reduction. The *x*-axis refers to the trait-like levels (baseline levels before the manipulation). The different lines indicate the state-like levels (amount of change that occurred from baseline to post-expectancy manipulation before the start of treatment). The black line indicates increase in activation (one SD above no change in activation), the red line refers to no change in activation (about 14% of the patients showed less than one standard deviation away from 0 in the amount of change), and the green line refers to decrease in activation (one SD below no change in activation). The error bars represent the 95% confidence intervals.

## Data Availability

Data is available upon request by email from the last author (brr8@cumc.columbia.edu).

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
