# Peer review of "Proof of Concept of the Contribution of the Interaction between Trait-like and State-like Effects in Identifying Individual-Specific Mechanisms of Action in Biological Psychiatry"

_jpm, 2022, doi:10.3390/jpm12081197_

Round 1

Reviewer 1 Report

The authors introduced an interesting framework for delineating the different components contributing to the effects we measure in participants/patients. To elucidate information on this framework, the authors used an example from one of their published studies.

The paper is well written, straightforward, and clear. I however have some few concerns:

Major comments

1. I would appreciate some original figure to better conceptualize this model of discriminating between state-like and trait-like effects in a RCT or any other experiment. What I find however is a review of what the authors called SL and TL from some previous studies. The authors explained using many examples from MDD to Schizophrenia, but just listing what the SL/TL were in these studies is not the same as a framework to discriminate TL/SL. This makes it a bit confusing if you consider the fact that the whole paper is about introducing a framework for studying TL/SL effects in clinical studies. The question then is, if I am to undertake an experiment today to study the effects of drug A on blood pressure for example, how to I design my experiment considering TL/SL framework? What are the general points to consider, and how do I proceed once those points are considered?. Perhaps a flowchart of some basic rules in the framework?.

2. I find half the citations are self-citations. There is nothing inherently wrong here, just curious if there are other studies where the discrimination of participant characteristics, and treatment characteristics was discussed into detail.

3. Was there any differences between the different times TL, and SL were measured?. A main effect of time?.

Minor comments

1. Please consider writing fully what the three way interaction was in the description of the results? SL, TL, and time?

2. In the introduction, the sentence ‘An early example is a classical editorial by Curfman (1993)…’ the word ‘which’ is missing.

3. In the first study you described (Rutherford et al 2017), the subsection ‘Design and Methods’ you wrote …. “At the Week 0 visit, post-randomization outcome expectancy and depression scores were measured, with participants having this additional information”. The last part is a bit confusing. If participants provided this information, it naturally means they have it. Please consider rewriting the sentence. OR, did you mean, the participants still had this information because it was not expected to dissipate?.

Author Response

We would like to thank the reviewer for the helpful feedback and suggestions. 

The review was helpful in pointing out weaknesses in our manuscript, and we are grateful for the opportunity to incorporate the requested changes. We believe that in its current form the manuscript is much improved, and hope that you will find it suitable for publication.

  1. I would appreciate some original figure to better conceptualize this model of discriminating between state-like and trait-like effects in a RCT or any other experiment. What I find however is a review of what the authors called SL and TL from some previous studies. The authors explained using many examples from MDD to Schizophrenia, but just listing what the SL/TL were in these studies is not the same as a framework to discriminate TL/SL. This makes it a bit confusing if you consider the fact that the whole paper is about introducing a framework for studying TL/SL effects in clinical studies. The question then is, if I am to undertake an experiment today to study the effects of drug A on blood pressure for example, how to I design my experiment considering TL/SL framework? What are the general points to consider, and how do I proceed once those points are considered?. Perhaps a flowchart of some basic rules in the framework?.

We clarified the implications of the proposed framework for designing future randomized control trials in the area of precision medicine. Specifically, we explained that at the design level, repeated measures are needed, at least at three time points, preferably more. The exact number of assessment points required may differ based on the nature of the SL and TL components of the construct. To accurately capture the SL component, the frequency at which the therapeutic construct of interest should be sampled depends on the rate of change that is expected in that construct. The number of time points required and the required time resolution (moments in a session, segments of a few minutes in each session, between-sessions assessment, etc.) needed to accurately capture the SL component of a given construct depend on the nature of the construct under investigation. Specifically, if the rate of change is slower, less frequent assessments are needed, but if the rate of change in the construct of interest is high, session-by-session assessment, or even within-session assessment is necessary. For constructs showing a slow rate of change, less frequent assessment may be more adequate. If the construct of interest is expected to change in the time that elapses between the therapy sessions and not only during the sessions themselves, repeated measures using experience sampling should be considered as well.

To accurately capture the TL component of each construct, it is necessary to also take into account the nature of the construct. The most important question to ask is whether the TL component has a temporal pattern. Some traits have no temporal pattern (are not expected to change over time), and can be accurately estimated based on a single assessment, as they have no potential to change and there is no error in their estimation. Some demographic variables are of this type. Other constructs have no temporal pattern, but their assessment involves errors, and therefore multiple assessments are needed to accurately estimate the TL component of the construct; some intelligence measures are of this type. Other TL constructs do have a temporal pattern, that is, they are dynamic trait characteristics, and taking into account the temporal dynamics is critical when estimating these TL components. In these cases, it is important to capture the dynamic TL patterns of the construct before the start of an intervention (Shahar, Bar-Kalifa, & Alon, 2017). When a TL dynamic is present in the construct of interest, the SL change may manifest as a change in the construct mean or slope above the TL pattern, as a change in the TL pattern, or both.

  1. I find half the citations are self-citations. There is nothing inherently wrong here, just curious if there are other studies where the discrimination of participant characteristics, and treatment characteristics was discussed into detail.

We reduced the number of self-citations. We also added more citations from other research groups.

  1. Was there any differences between the different times TL, and SL were measured?. A main effect of time?.

We clarified that for all patients included in the analyses, two time points were available, one at baseline and the other post-treatment. We referred to the baseline pre-manipulation levels as the “TL component,” and to changes from baseline pre-manipulation to post-manipulation as “SL changes.” Because the study was based on a pre-post design, as commonly used in neurobiological studies in psychiatry, it was not possible to test for a main effect of time. We highlighted that repeated measurement designs with a high number of data points are preferable and should be the focus of future research.

Minor comments

  1. Please consider writing fully what the three way interaction was in the description of the results? SL, TL, and time?

We clarified that the 3-way interaction was of SL, TL, and time.

  1. In the introduction, the sentence ‘An early example is a classical editorial by Curfman (1993)…’ the word ‘which’ is missing.

Corrected.

  1. In the first study you described (Rutherford et al 2017), the subsection ‘Design and Methods’ you wrote …. “At the Week 0 visit, post-randomization outcome expectancy and depression scores were measured, with participants having this additional information”. The last part is a bit confusing. If participants provided this information, it naturally means they have it. Please consider rewriting the sentence. OR, did you mean, the participants still had this information because it was not expected to dissipate?.

We clarified the design used by Rutherford et al. (2017). Specifically, the authors manipulated expectancy through instructions to participants about the probability of receiving active medication compared to receiving placebo. Participants at baseline had what they perceived to be a 75% probability of receiving active antidepressant medication. Pre-randomization expectancy was measured with participants having this knowledge. Then, they were randomly assigned either to the placebo-controlled condition (50% chance of receiving active treatment) or to the open condition (100% chance of receiving active treatment), and informed of the results of the randomization. Post-randomization expectancy was measured with participants having this additional information. Those in the placebo-controlled condition were randomly assigned to medication or placebo. The second randomization within the placebo-controlled group was blinded, and neither participants nor outcome assessors were aware of the randomization schedule or the specific treatment assignment to medication or placebo.

Reviewer 2 Report

.

Author Response

I would have appreciated that the authors had added line numbers in the document to make the review easier.

We added line numbers to the document.

Title: I don't like a title with a gerund and a complex term Disentangling that not all readers know and it is not essential.

The manuscript focuses on the importance of disentangling trait-like and state-like components. The title, accordingly, refers to disentangling trait-like and state-like components. The entire manuscript explains what it means to disentangle the two and how this can be accomplished. To address the reviewer’s concern, we replaced the title with the following one: Proof of concept of the contribution of the interaction between trait-like and state-like effects in identifying individual-specific mechanisms of action in biological psychiatry.

Keywords: Among the keywords only moderator and precision medicine appear in the Mesh, the authors must choose their keywords among those that best match in the Medical Subjet Headings.

The list of keywords is as follows: Mechanisms of action; between-individuals variance; within-individual variance; precision medicine; mediators; moderators

I think the nomenclature used to differentiate the TL/SL differentiation or interaction may be valid but needs to be explained.

We further clarified how and why it is important to differentiate between the trait-like and state-like components of each mechanism of action.

There are many layout errors (e.g: the image in figure 1 is not centered) and typos

Corrected.

Discussion: we recommend an interdisciplinary collaboration between clinical researchers, data science experts, and clinicians in identifying candidate TL components that may moderate treatment response... should not be the conclusion as it is a is a vague and imprecise generality.

We clarified these recommendations and how they are connected to the framework we propose in this manuscript.

Figures: All the figures seem to be uninformative.

We replaced Figure 1 with a more informative figure. We clarified the information presented in Figures 2 and 3.

I understand the article but, as I am not a native English speaker; a priori I would think that the authors speak English better than I do but I find many strange expressions that are not familiar to me from standard scientific English so I don't feel qualified to judge about the English language and style. There are expressions like: ”In such fashion…” (Conclusions in the abstract) or “putative therapeutic mechanism…” (Introduction) that sound strange to the non-native reader.

The manuscript has been proofread to correct potential mistakes.

I have detected 12 self-citations, and it seems to be too much for me.

We reduced the number of self-citations. We also added more citations from other research groups.

Reviewer 3 Report

This paper describes an innovative approach to personalized medicine in the specific context of pharmacological treatments in psychiatry - the utility of state-like and trait-like effects, and the interaction between them, in predicting treatment effects. In the specific case discussed by the authors, patients were assigned to either a standard antidepressant drug (citalopram) or placebo, and were given two neuroimaging tasks (amygdala activation in relation to the recognition of faces, and pallidum activation in relation to monetary rewards or penalties). The authors were able to disentangle state-like and trait-like effects in the study participants, and found that the interaction between the two played an important role in predicting the response to treatment.

I read this paper with considerable interest and agree with the authors' contention that this method of analysis represents an approach that is both innovative and valid.

I found no significant flaws either in the conceptual arguments presented in this paper or in the study methodology. The current version of the manuscript could be improved further by attention to the following, relatively minor, concerns:

1. As the authors correctly point out, the term "trait" or "trait-like" has been used in at least two ways in the literature: (a) as long-term, enduring and "static" psychological or biological factors, and (b) as baseline values of a parameter prior to the introduction of any form of treatment. The current paper seems to use definition (b); this could be made a little clearer when discussing the study methodology and results.

2. The high attrition in the trial could be considered a limitation to a certain extent. While this paper is primarily concerned with the presentation of a key preliminary finding, it is possible that the study findings could have been altered if data was available for a larger sample. (More specifically, was there any difference in baseline fMRI parameters between those whose data was complete and analyzed, and those who whose was incomplete? If so, this could confound the interpretation of the results presented here.)

3. Several other baseline characteristics (which could be considered "trait" factors) have been associated with antidepressant response; the most consistently documented factor of this sort is childhood adversity (cf. Zisook et al., 2019; Medeiros et al., 2021). Was any information on baseline parameters of this sort available for analysis in the study participants? (This would not be considered a limitation of the current study, but it could be highlighted as an avenue for further research; there may be several TL/SL interactions involved in the prediction of antidepressant response.)

Author Response

We would like to thank the reviewer for the helpful feedback and suggestions. 

The review was helpful in pointing out weaknesses in our manuscript, and we are grateful for the opportunity to incorporate the requested changes. We believe that in its current form the manuscript is much improved, and hope that you will find it suitable for publication.

  1. As the authors correctly point out, the term "trait" or "trait-like" has been used in at least two ways in the literature: (a) as long-term, enduring and "static" psychological or biological factors, and (b) as baseline values of a parameter prior to the introduction of any form of treatment. The current paper seems to use definition (b); this could be made a little clearer when discussing the study methodology and results.

We made this important point clearer in the text.

  1. The high attrition in the trial could be considered a limitation to a certain extent. While this paper is primarily concerned with the presentation of a key preliminary finding, it is possible that the study findings could have been altered if data was available for a larger sample. (More specifically, was there any difference in baseline fMRI parameters between those whose data was complete and analyzed, and those who whose was incomplete? If so, this could confound the interpretation of the results presented here.)

We added that of the patients participating in the RCT, 23 met imaging criteria (no MRI-contraindications, etc.), and formed the effective sample for this secondary analysis. Of these patients, 9 were randomized to the Open group and 14 to the Placebo-controlled group (11 received medication and 3 received placebo). No significant differences in demographic data or baseline clinical characteristics were found between participants who were and were not scanned, or between participants in the Placebo-controlled and Open groups.

  1. Several other baseline characteristics (which could be considered "trait" factors) have been associated with antidepressant response; the most consistently documented factor of this sort is childhood adversity (cf. Zisook et al., 2019; Medeiros et al., 2021). Was any information on baseline parameters of this sort available for analysis in the study participants? (This would not be considered a limitation of the current study, but it could be highlighted as an avenue for further research; there may be several TL/SL interactions involved in the prediction of antidepressant response.)

We agree with the reviewer that it can be of great interest to test the ability of childhood adversity to serve as a potential moderator. Unfortunately, this variable was not collected for the included patients. We noted that this can be of special interest for future research.